# A highly efficient and accurate method of detecting and subtyping Influenza A pdm H1N1 and H3N2 viruses with newly emerging mutations in the matrix gene in Eastern Taiwan

Hui-Hua Yang[1,2,3⊚], I-Tsong Huang[3,4⊚], Ren-Chieh Wu[5,6,7], Li-Kuang Chen[3,4,5,6,7,8]*

1 Bioinnovation Center, Buddhist Tzu Chi Medical Foundation, Hualien, Taiwan, 2 Department of Medical Research, Hualien Tzu Chi Hospital, Buddhist Tzu Chi Medical Foundation, Hualien, Taiwan, 3 Taiwan CDC Collaborating Laboratories of Virology, Hualien Tzu Chi Hospital, Buddhist Tzu Chi Medical Foundation, Hualien, Taiwan, 4 Department of Laboratory Medicine, Hualien Tzu Chi Hospital, Buddhist Tzu Chi Medical Foundation, Hualien, Taiwan, 5 Branch of Clinical Pathology, Department of Laboratory Medicine, Hualien Tzu Chi Hospital, Buddhist Tzu Chi Medical Foundation, Hualien, Taiwan, 6 Department of Emergency Medicine, Hualien Tzu Chi Hospital, Buddhist Tzu Chi Medical Foundation, Hualien, Taiwan, 7 PhD Program in Pharmacology and Toxicology, Tzu Chi University, Hualien, Taiwan, 8 Institute of Medical Sciences, Department of Laboratory Diagnostic, College of Medicine, Tzu Chi University, Hualien, Taiwan

⊚ These authors contributed equally to this work.
* likuangchen@gmail.com

**Data Availability Statement:** All relevant data are within the paper and its Supporting Information files.

## Abstract

The rapid identification of Influenza A virus and its variants, which cause severe respiratory diseases, is imperative to providing timely treatment and improving patient outcomes. Conventionally, two separate assays (total test duration of up to 6 h) are required to initially differentiate Influenza A and B viruses and subsequently distinguish the pdm H1N1 and H3N2 serotypes of Influenza A virus. In this study, we developed a multiplex real-time RT-PCR method for simultaneously detecting Influenza A and B viruses and subtyping Influenza A virus, with a substantially reduced test duration. Clinical specimens from hospitalized patients and outpatients with influenza-like symptoms in Eastern Taiwan were collected between 2011 and 2015, transported to Hualien Tzu Chi Hospital, and analyzed. Conventional RT-PCR was used to subtype the isolated Influenza A viruses. Thereafter, for rapid identification, the multiplex real-time RT-PCR method was developed and applied to identify the conserved regions that aligned with the available primers and probes. Accordingly, a multiplex RT-PCR assay with three groups of primers and probes (MAF and MAR primers and MA probe; InfAF and InfAR primers and InfA probe; and MBF and MBR primers and MB probe) was established to distinguish these viruses in the same reaction. Thus, with this multiplex RT-PCR assay, Influenza B, Influenza A pdm H1N1, and Influenza A H3N2 viruses were accurately detected and differentiated within only 2.5 h. This multiplex RT-PCR assay showed similar analytical sensitivity to the conventional singleplex assay. Further, the phylogenetic analyses of our samples revealed that the characteristics of these viruses were different from those reported previously using samples collected during 2012–

**Funding:** This work was supported by Grants from the Centers for Disease Control, R.O.C (Taiwan). (HZ099103, HZ100102, HZ101060, CH102055, CW103035). The funders had no role in study design, data collection and analysis, decision to publish, or preparation of the manuscript.

**Competing interests:** The authors have declared that no competing interests exist.

2013. In conclusion, we developed a multiplex real-time RT-PCR method for highly efficient and accurate detection and differentiation of Influenza A and B viruses and subtyping Influenza A virus with a substantially reduced test duration for diagnosis.

## Introduction

Different Influenza A serotypes and virus strains have caused several pandemics over the past 100 years [1, 2]. Notably, the pdm H1N1 and H3N2 serotypes of Influenza A virus can infect humans and cause seasonal flu. Influenza A virus infections can also cause severe respiratory diseases, leading to considerable complications and mortality in immunocompromised patient groups, the elderly, and children. Pandemics caused by Influenza A viruses affect public health, resulting in lost work time as well as huge economic losses [3–5]. Thus, the rapid and immediate identification of various Influenza A virus serotypes is necessary to provide clinical and timely medication to patients [6]. Furthermore, rapid detection methods may provide information regarding the influenza virus vaccine that needs to be used [7, 8].

Real-time reverse transcription polymerase chain reaction (real-time RT-PCR) is the most rapid, effective [8, 9], and extensively used method of detecting Influenza A virus with high sensitivity and specificity. It has also become the primary technique for Influenza A virus detection [6, 8]. The matrix protein gene (M gene) of Influenza A viruses showes high conservation and stability and is used as the target gene for real-time RT-PCR [10]. Given that Influenza A viruses undergo genetic mutations frequently, the identification of mutated Influenza A viruses can provide timely information on clinically suitable medications [10–12]. Influenza A viruses can cause pandemics and create a giant loophole in public health, which involves disease prevention and control [13]. Reportedly, the pdm H1N1 A and H7N9A serotypes of Influenza A viruses caused severe epidemics in humans in 2009 and 2013, respectively. Thus, in 2009, the World Health Organization (WHO) recommended primers and probes (MAF, MAR, and MA probe) that can be used for the rapid detection of the pdm H1N1A serotype [10]. Presently, these primers and probes are widely used by many diagnostic laboratories [10].

After 2011, the H3N2 influenza virus, with a mutated M gene emerged. This M-mutant H3N2 virus can also be detected using the WHO-recommended primers and probes [10], but with less sensitivity, which could easily lead to false-negative results in influenza virus testing laboratories [1]. Comparative analysis of the M gene of Influenza A H3N2 virus has indicated that mutations between positions 167 and 274 may reduce the degree of MAF-primer pairing adhesion and further reduce the sensitivity of the detection method. Additionally, the primers and probes that are currently used to detect Influenza A viruses were designed to target an M protein gene sequence with a small chance of detecting gene mutation. However, mutation in the M protein gene results in the failure of primers and probes to bind to the target sequence. Therefore, given that the primers and probes (MAF, MAR, and MA probe) recommended by the WHO have become insensitive in detecting Influenza A virus H3N2 strains after 2011, an improved real-time RT-PCR test is needed to ensure accurate diagnosis [1, 10].

To detect these viruses, a two-step procedure with two separate assays has been developed and extensively used (**Fig 1A**). Using the primers and probes (MAF, MAR, and MA probe; and MBF, MBR, and MB probe) recommended by WHO for detecting Influenza A and B viruses, the real-time RT-PCR assay usually takes 2.5 to 3 h for the first step [10]. Using this traditional RT-PCR assay, an additional 2.5 to 3 h is often needed to differentiate Influenza A virus pdm H1N1 or H3N2 serotypes. Thus, a total of 5–6 h is often needed to differentiate

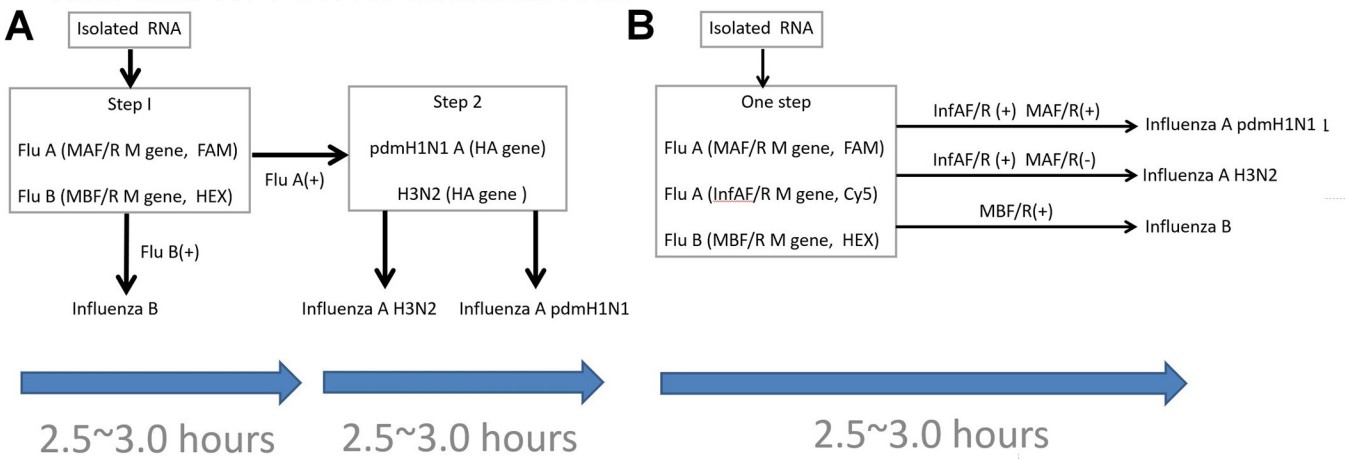

**Fig 1. Real-time multiplex RT-PCR and conventional singleplex RT-PCR assays for the detection of Influenza B, Influenza A pdm H1N1, and Influenza A H3N2 viruses. A.** Conventional two-step procedure with two separate assays for detection. **B.** One-step detection procedure using the multiplex RT-PCR assay developed in this study. The primers and probes used for detection are indicated. Flu, influenza; RNA, ribonucleic acid; MA, InfA and MB, three types of primers; F/R, forward/reverse, (+) and (-), positive and negative, respectively; HA, hemagglutinin gene; FAM, HEX, and Cy5, different Taqman probes for the assays.

pdm H1N1 or H3N2 serotype viruses. This implies that improvements of the real-time RT-PCR method are needed to shorten the time required for the identification of the pdm H1N1 or H3N2 serotype Influenza A viruses.

In this study, we aimed to develop a novel multiplex RT-PCR assay (one-step procedure, Fig 1B) with three groups of primers and probes (MAF, MAR, and MA; InfAF, InfAR, and InfA; and MBF, MBR, and MB probe) for distinguishing Influenza B, Influenza A pdm H1N1, and Influenza A H3N2 viruses in the same reaction. We postulated that this novel assay would offer the possibility to effectively detect Influenza B, Influenza A pdm H1N1, and Influenza A H3N2 viruses within a substantially reduced time. In this study, we also report the results of the phylogenetic analyses of Influenza A pdm H1N1 and H3N2 viruses associated with the pandemic that occurred during 2012–2013 in Taiwan [14]. Since then, the changes in phylogenetic characteristics of these viruses have not yet been investigated. Therefore, in this study, we further performed phylogenetic analyses for the Influenza A pdm H1N1 and H3N2 viruses that caused the 2011–2015 pandemic.

## Materials and methods

### Ethics statement

Ethical approval for this study was granted by the Research Ethics Committee of Hualien Tzu Chi Hospital, Buddhist Tzu Chi Medical Foundation (Approval number IRB111-056-B). Informed consent was waived according to the institutional guidelines. Influenza virus is a serious infectious disease in Taiwan. The ethical standards and laboratory diagnosis for the handling of the Influenza A virus samples were based on the regulations of the Taiwan Center for Disease Control (CDC).

### Clinical specimen collection and viral RNA extraction

Between 2011 and 2015, a total of 5,709 clinical specimens were collected in the form of throat swabs from hospitalized patients, outpatients with influenza-like illness in the community, and patients with complicated influenza infections in Eastern Taiwan [15]. These specimens were

**Table 1. Primer and probe sequences for the real-time RT-PCR for the detection of Influenza A and B strains.**

| Name | Primer Sequence (5′→3′) | Labels (5′–3′) | Target gene position | Accession number | References |
|------|------------------------|----------------|---------------------|------------------|------------|
| MAF | AAG ACC AAT CCT GTC ACC TCT GA | | 145–166 | X08092 | (10) |
| MAR | CAA AGC GTC TAC GCT GCA GTC C | | 217–238 | | |
| MA probe | TTT GTG TTC ACG CTC ACC GT | FAM, IBFQ | 184–201 | | |
| InfAF | GAC CRA TCC TGT CAC CTC TGA C | | 146–167 | X08092 | CDC US (17) |
| InfAR | AGG GCA TTY TGG ACA AAK CGT CTA | | 228–251 | | |
| InfA probe | TGC AGT CCT CGC TCA CTG GGC ACG | Cy5, IBFQ | 201–224 | | |
| MBF | GAG ACA CAA TTG CCT ACC TGC TT | | 17–39 | CY018758 | (10) |
| MBR | TTC TTT CCC ACC GAA CCA AC | | 92–110 | | |
| MB probe | AGA TGG AGA AGG CAA AGC AGA ACT AGC | HEX, IBFQ | 40–77 | | |

MA, InfA and MB, three types of primers; F/R, forward/reverse; FAM, HEX, and Cy5, different Taqman probes for assays; IBFQ, Fluorescent Quencher.

then transported to the Taiwan CDC Collaborating Laboratories of Virology at Hualien Tzu Chi General Hospital [1, 2, 15]. After nucleic acid extraction from the samples by using an automated TANBead system (Taiwan Advanced Nanotech, Inc., Taiwan) or the QIAamp® Viral RNA Kit (Qiagen, Hilden, Germany), the hemagglutinin (HA) types were identified via traditional RT-PCR. Thereafter, the nucleic acid samples were subjected to multiplex real-time RT-PCR. All the experiments involving pathogenic viruses were conducted at biosafety level 2 containment [13].

## Primers and probes for real-time RT-PCR and conventional RT-PCR

The primers and probes used for the detection of Influenza A and B viruses were those recommended by the USA CDC [16] or WHO [10], as shown in Table 1. The first primers and probe (MAF, MAR, and MA probe) were used to detect the Influenza A strains pdm H1N1, H3N2, and other strains identified since 2009. The second primers and probe (InfAF, InfAR, and InfA probe) were used for the routine detection of mutant Influenza A strains H3N2, pdm H1N1, and other strains identified since 2011 [10, 14]. The real-time RT-PCR probes were labeled with a reporter dye at the 5′-end and a quencher at the 3′-end. Table 1 shows the nucleotide sequences and labels used among the primers and probes.

Multiplex real-time assays were performed in a one-step reaction using a Qiagen one-step RT-PCR kit (Qiagen, Hilden, Germany) with 5 µL of the viral RNA extracted from the clinical samples [14]. Each reaction included primers (0.5 µM), probe (0.1 µM), deoxyribonucleotide triphosphate mixture (0.5 mM), and one-step enzyme mixture (1 µL) in a final volume of 25 µL. A Bio-Rad iQ5 instrument (Bio-Rad Diagnostics, Hercules, CA, USA) was used for amplification and detection. The cycle conditions were as follows: 30 min at 50˚C, followed by 15 min at 95˚C and then 45 cycles of 15 s at 95˚C and 1 min at 60˚C [13, 16]. The stimulated fluorescence signals of the probes were read at the end of every cycle, and the raw data were calculated to analyze the PCR product amplification curve.

The primers used in the RT-PCR assays to detect the strain subtypes (Table 2) were those recommended by the Taiwan CDC [16]. Further, RT-PCR amplification was performed in a reaction volume of 25 µL using a Qiagen one-step RT-PCR kit (Qiagen, Hilden, Germany) and a Bio-Rad iQ5 instrument (Bio-Rad Diagnostics, Hercules, CA, USA) according to the manufacturer's protocol. The reaction mixture contained 0.2 µM of each primer, 0.5 mM of deoxyribonucleotide triphosphate mixture, and 1 µL of one-step enzyme mixture in a final volume of 25 µL. Further, we used RNA extracted from patients with Influenza A/New Caledonia/20/99(H1N1), Influenza A/Wisconsin/67/05(H3N2), and Influenza B Florida/07/04 as

**Table 2. Primer sequences of the RT-PCR for HA subtyping and HA gene and MA gene-sequencing reaction.**

| Name | Primer Sequence (5′→3′) | Target gene position | Amplicon | Accession number | References |
|------|-------------------------|---------------------|----------|------------------|------------|
| H3-F | ACT ATC ATT GCT TTG AGC | 7–24 | | CY045732 | TWCDC |
| H3-R | TGG CAT AGT CAC GTT CAG | 538–555 | 550bp | | |
| pdmH1F | TGC ATT TGG GTA AAT GTA ACA TTG | 216–239 | | CY060558 | TWCDC |
| pdmH1R | AAT GTA GGA TTT RCT GAK CTT TGG | 526–549 | 340bp | | |
| M-1 | AGC AAA AGC AGG TAG ATA TT | 1–20 | | KY049991 | (16) |
| M-1027R | AGT AGA AAC AAG GTA GTT TTT | 1007–1027 | 1027bp | | |
| HA-1 | AGCAAAAGCAGGGGAAAATA | 1–20 | | KX136363 | (16) |
| HA-1778R | AGTAGAAACAAGGGTGTTTT | 1779–1759 | 1779bp | | |

H3, pdmH1, M and HA, four types of primers; F/R, forward/reverse.

positive controls, while sterile water served as the negative control in each run. Specific fragments were amplified under the following conditions: 30 min at 50°C; 10 min at 95°C; 40 cycles of 30 s at 94°C, 45 s at 55°C, and 1 min at 72°C; and one cycle of 10 min at 72°C. Further, the amplification products were separated via electrophoresis. The gels were stained with ethidium bromide, and the amplicons were visualized using an ultraviolet transilluminator (UVP BioImaging Systems, Cambridge, UK). The molecular weights of the PCR products (Table 2) were calculated using LabWorks™ Analysis Software UVP version 3.0.02.00 (Solutions for Science of Life).

## Cell culture and virus identification

Madin–Darby canine kidney (MDCK) cells were prepared and provided by the Taiwan CDC. The cells were maintained in Dulbecco's modified Eagle's medium supplemented with L-glutamine (1 mM; Gibco), HEPES (1 mM; Gibco), and 10% heat-inactivated fetal bovine serum at 37°C with 5% $CO_2$. The clinical specimens were sterilized using a 0.45-μm syringe filter and propagated into the MDCK cells for virus culture and nucleic acid extraction, after which the viruses were harvested from the MDCK cells and tested for further identification [9]. The virus serotypes were identified using an indirect immunofluorescence assay, and the HA subtypes of Influenza A viruses were detected via conventional RT-PCR.

## Viral cDNA sequencing

Viral cDNA sequencing of the Influenza A viruses was performed using freshly cultured isolates from the clinical specimens or nucleic acids from clinical samples. To analyze the phylogenetic relationships between the HA of the isolated pdm H1N1 and H3N2 serotypes, the HA and M genes were first examined via conventional RT-PCR using primers and a protocol as previously described [16, 17]. Viral RNA was extracted from the freshly cultured isolates using an automated TANBead system (Taiwan Advanced Nanotech, Inc., Taiwan) or the QIAamp® Viral RNA Kit (Qiagen, Hilden, Germany). Further, the RT-PCR products were purified using a High Pure PCR Product Purification Kit (Roche Diagnostics, Mannheim, Germany) and cloned into the TA plasmid vector using a pGEM®-T Easy Vector Systems kit (Promega, Madison, WI, USA).

Plasmid DNA sequencing for the entire length of the cloned gene was performed using an ABI Prism 3730 DNA sequencer (Applied Biosystems, Foster City, CA, USA). Two or three different plasmid DNAs of the same Influenza A virus isolate were sequenced to determine the accuracy of the gene sequencing. The gene sequences were then assembled using the Basic Local Alignment Search Tool from the National Center for Biotechnology Information [18].

## Genetic analyses of Influenza A viruses

The assembled nucleotide sequences of Influenza A viruses were aligned using ClustalW software. Further, phylogenetic and molecular analyses were performed using MEGA version 10.0 and BioEdit software (http://www.mbio.ncsu.edu/BioEdit/bioedit.html). The HA and M genes of the isolated Influenza A viruses were compared with those from the GenBank database, and alignments were performed using the ClustalW multiple alignment tool [19, 20]. A phylogenetic tree was also constructed using the neighbor-joining method, and 1,000 bootstrap replications were performed to evaluate its reliability [21]. The phylogenetic relationships among the studied and reference sequences were also evaluated to estimate the node reliability of the phylogenetic trees constructed using two methods with 1,000 bootstrap replicates, and bootstrap support values > 75 were considered significant [19].

## Statistical analysis

Continuous variables were compared via the independent sample $t$-test and presented as mean ± standard deviation (SD). The correlations between data obtained from multiplex assays and singleplex assays were analyzed by performing Pearson correlation analysis, and statistical significance was set at $p < 0.05$. Further, all the statistical analyses were performed using SPSS software version 23.0 (SPSS Inc., Chicago, IL, USA) for Windows.

# Results

## Real-time RT-PCR detection of Influenza A viruses

Among the 5,709 samples collected, 558, 401, and 370 cases were positive for Influenza A H3N2, Influenza A pdm H1N1, and Influenza B viruses based on real-time RT-PCR. Further, we analyzed isolated RNA from these samples using the multiplex real-time RT-PCR method developed in this study, which included three groups of primers and probes for Influenza A and B detection in the same PCR tube (Fig 1B). Fig 2 shows two representative PCR data from our one-step multiplex assay. For the sample with data showing InfAF/R (+) and MAF/R(+) (Fig 2A), the case was diagnosed as Influenza A pdm H1N1 virus. For the sample with data showing InfAF/R (+) and MAF/R(-) (Fig 2B), the case was diagnosed as Influenza A H3N2

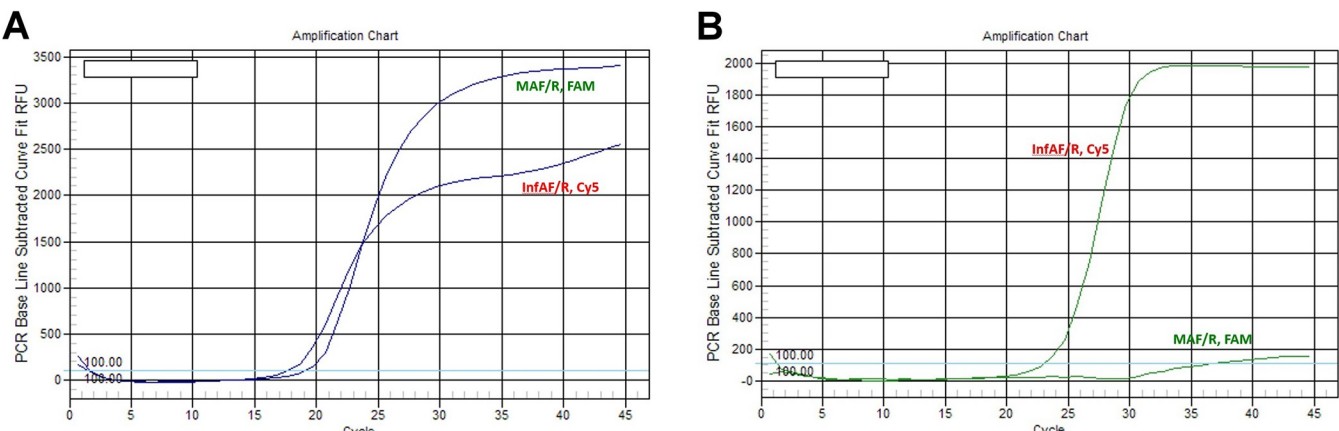

**Fig 2. Representative real-time multiplex RT-PCR curves for the detection and differentiation of Influenza A pdm H1N1 and H3N2 viruses. A.** Data showing InfAF/R (+) and MAF/R (+) for the diagnosis of Influenza A pdm H1N1 virus. **B.** Data showing InfAF/R (+) and MAF/R (-) for the diagnosis of Influenza A H3N2 virus. The horizontal blue lines represent the threshold cycle ($C_T$) values. MA and InfA, two types of primers; F/R, forward/reverse; FAM and Cy5, different Taqman probes for the assays.

**Table 3. Analytical sensitivity of the singleplex and multiplex assays analyzed by standard curves of $C_T$ values using virus RNA.**

| Standard concentration | | Singleplex assay | | | Multiplex assay | | |
|---|---|---|---|---|---|---|---|
| Copies/μl | Log$_{10}$ Conc. | MAF/R | MBF/R | InfAF/R | MAF/R | MBF/R | InfAF/R |
| $10^5$ | 5 | 20.11±0.58 | 21.91±0.30 | 18.28±0.36 | 21.47±0.49 | 21.77±0.32 | 18.94±0.39 |
| $10^4$ | 4 | 23.20±0.51 | 25.31±0.31 | 21.55±0.36 | 25.00±0.49 | 25.16±0.29 | 22.47±0.37 |
| $10^3$ | 3 | 26.93±0.71 | 28.67±0.33 | 25.08±0.35 | 28.61±0.46 | 28.57±0.32 | 26.08±0.37 |
| $10^2$ | 2 | 30.23±0.51 | 32.15±0.32 | 28.58±0.33 | 32.33±0.48 | 32.11±0.28 | 29.81±0.43 |
| $10^1$ | 1 | 33.10±0.51 | 35.41±0.38 | 32.11±0.34 | 36.00±0.46 | 36.00±0.28 | 33.50±0.36 |
| $10^\circ$ | 0 | 36.85±0.50 | 38.70±0.71 | 35.42±0.38 | 39.39±0.50 | 38.27±0.60 | 37.25±0.36 |
| R | | 0.9995 | 1.0000 | 0.9999 | 0.9999 | 0.9984 | 0.9999 |
| Slope | | -3.3343 | -3.3637 | -3.4537 | -3.6091 | -3.3874 | -3.6677 |
| E | | 99.4913 | 98.3145 | 94.8064 | 89.2716 | 97.3543 | 87.3718 |

Data ($C_T$ values) are presented as the mean ± standard deviation. The correlations of data obtained from multiplex assays and singleplex assays were analyzed by Pearson correlation. Data obtained from multiplex assays and singleplex assays were compared using independent sample t-test and no significances were detected. R, correlation coefficient; E, amplification efficiencies of the assays.

virus. In these two curves, the positive and negative findings were defined as curve responses above and below the threshold cycle ($C_T$) values. Further, the presence of Influenza A pdmH1N1 or Influenza A H3N2 viruses in each tested sample was confirmed by the HA assays. The threshold cycle ($C_T$) values of some representative tested samples diagnosed as positive for Influenza A pdmH1N1 and Influenza A H3N2 viruses are presented in S1 and S2 Tables (Supplemental Materials), respectively. Table 3 shows the analytical sensitivity of the multiplex assays developed in this study and the traditional singleplex assays. The $C_T$ values for each standard concentration obtained from the use of MAF/R, MBF/R, and InfAF/R in the multiplex assays did not significantly differ from those obtained in the singleplex assays. Further analyses of the standard curves of the $C_T$ values revealed that the correlation coefficients, slopes, and amplification efficiencies of these two types of assays were not significantly different.

## Nucleotide sequence alignments between the M gene sequences of Influenza A viruses and the primers and probes used in this study

The ClustalW multiple alignment of the M sequences of the Influenza A viruses and the primers and probes used in the real-time RT-PCR detection in this study are shown in **Fig 3**. Some mutations were found to be located between positions 167 and 274 (A/USA/AF1096/ 2007 (H3N2)), for which the primer and probe binding sites of the commonly used real-time RT-PCR assay primers and probes are designed.

The $C_T$ value obtained after the assay performed using the set of MAF and MAR primers, as well as the MA probe, was delayed for eight to nine cycles or undetectable for Influenza A H3N2 viruses. Position 212 of the M sequence changed from G to T (**Fig 3**), and possibly, was primarily responsible for these undetectable probe signals. Detection using this set of primers and probe was less sensitive because the forward primer, MAF, could not match the nucleotides of the M gene in the Influenza A H3N2 virus. This situation can be remedied by our second set of primers and probe (InfAF and InfAR primers, and InfA probe), which targeted a more conserved region of the M gene sequence of Influenza A H3N2 virus. Thus, the sensitivity of the real-time RT-PCR for the detection of Influenza A H3N2 viruses from 2011 was not reduced. This second set of primers and probe has been extensively used for the real-time RT-PCR of the M gene and is recommended by the WHO.

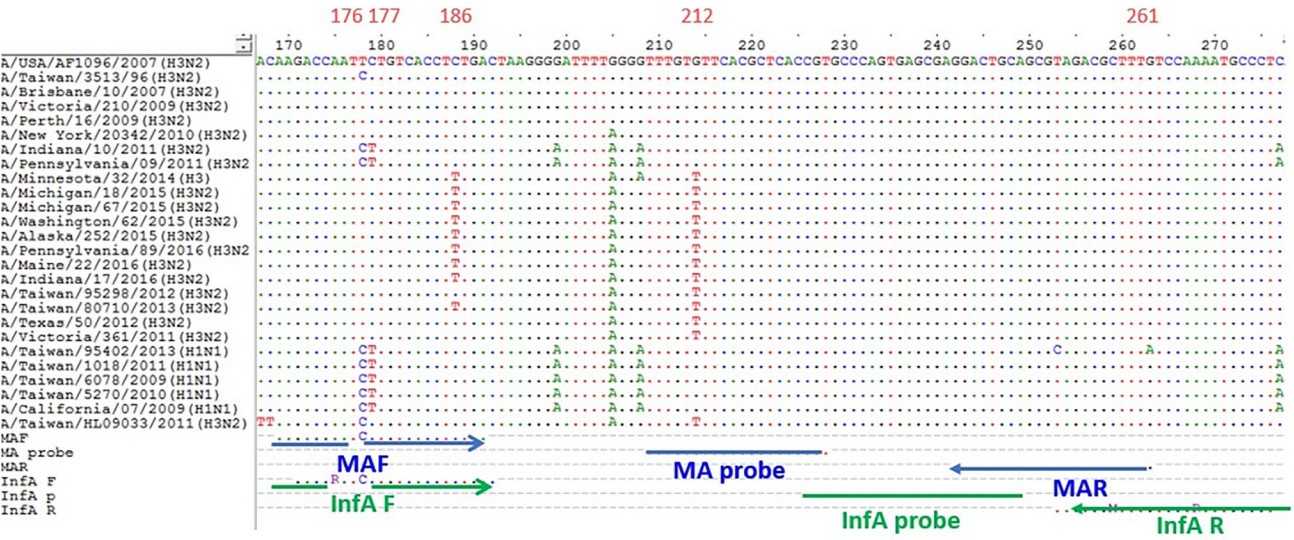

**Fig 3. Multiple alignments of M gene sequences from Influenza A pdmH1N1 and H3N2 viruses and the primer and probe sequences used in this study.** Multiple alignment results for the M gene sequences in different Influenza A virus HA subtypes and the two sets of primers and probes used for Influenza A virus detection via real-time RT-PCR. The M protein sequence of the A/USA/AF1096/2007 (H3N2) strain served as the reference sequence. The dashed lines indicate identical nucleotide bases with the other viruses, primers, and probes. Further, different nucleotide bases are directly indicated at positions 176, 177,186, 212, and 261. The positions show the mutant bases of Influenza A virus variants, which differed between the primer and probe sequences. The evolutionary clades of the viruses based on the HA sequence are also shown at the end of the virus name.

## Phylogenetic analysis of isolated Influenza A viruses

Among the isolated viruses, 44 samples of Influenza A pdm09 H1N1 virus and 50 samples of Influenza A H3N2 virus were randomly selected to determine their partial HA sequences. These samples, which served as representative isolates, were subjected to DNA sequencing for the detection of their partial-length HA sequences. The results obtained were then compared with those obtained for previous circulating viruses. Based on the HA phylogenetic analysis (**Fig 4A**), Influenza A pdm H1N1 viruses that re-emerged between 2011 and 2015 were classified as clades 11.1, 12.1, and 12.2, as well as the new clade 11.2, which branched from the previously predominant clades 11 and 12, comprising viruses that were in circulation from 2010 to 2011. Further, the Influenza A pdm H1N1 viruses isolated between January 2011 and July 2013 belonged to clade 12, and some of the pdm H1N1 viruses isolated from May to July 2013 belonged to clade 11.1, whereas the pdm H1N1 viruses isolated from December 2013 to November 2015 belonged to the new clade 11.2.

Based on the HA phylogenetic analysis of Influenza A H3N2 virus (**Fig 4B**), viruses that re-emerged within the 2011–2015 period were classified into two large clades, namely, TW2011–13 and the new clade TW2014–15. It was also observed that Influenza A H3N2 viruses isolated from January 2011 to July 2013 belonged to clade TW2011–13, whereas the H3N2 viruses isolated from December 2013 to December 2015 belonged to the new clade TW2014–15.

## Discussion

Quantitative real-time RT-PCR assays have been used to target the M gene of Influenza A viruses (MAF and MAR primers and MA probe) based on the assay developed by Ward et al. [10]. This assay has been used in many studies and for Influenza A virus detection; it is also recommended by the WHO owing to its high sensitivity, rapidity, and accuracy. However, it cannot be used to detect several Influenza A H3N2 virus variants from 2011, and in some

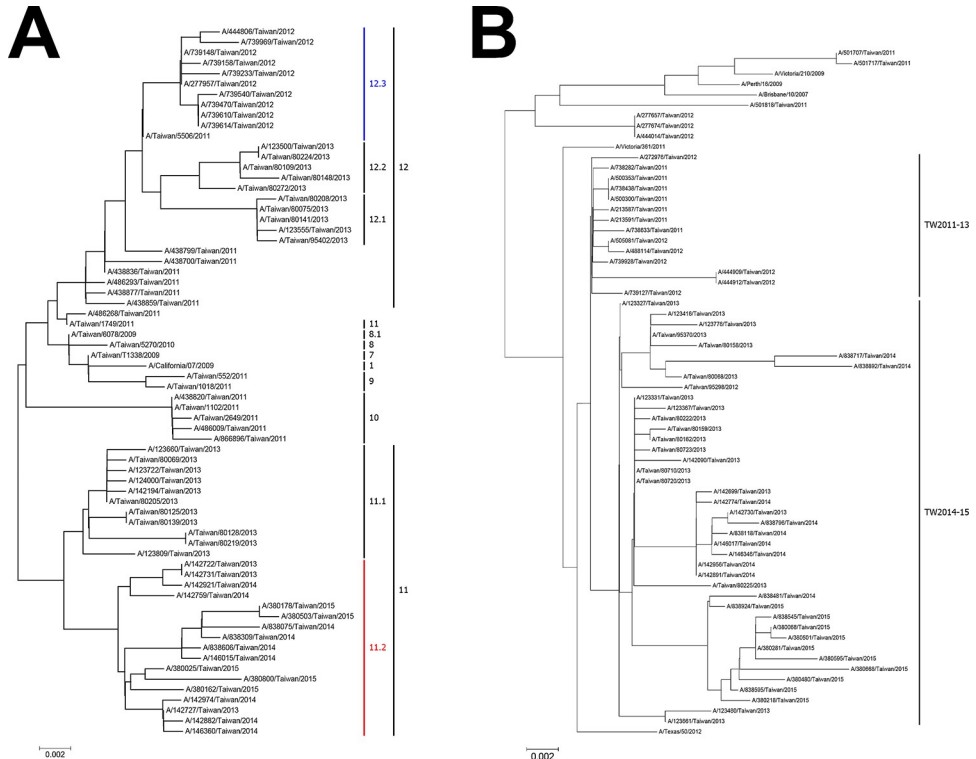

**Fig 4. Phylogenetic relationships of the full-length hemagglutinin gene.** A phylogenetic tree was constructed using the neighbor-joining method and MEGA software. Tree reliability was evaluated using 1,000 bootstrap replicates, and bootstrap support values greater than 75 are shown. **A.** Phylogenetic tree of Influenza A pdm H1N1 HA gene sequences. Clades 11.2 and 12.3 were differentiated from clades 11 and 12 in samples collected between 2011 to 2015. The new clade 12.3 was differentiated from clade 12 using samples collected between 2011 and 2012, and the new clade 11.2 was differentiated from clade 11 in samples from 2013 to 2015 in Eastern Taiwan. **B.** Phylogenetic tree of Influenza A virus H3N2 HA gene sequences. The new clade TW2014–2015 was differentiated from clade TW2012–2013 using samples from 2014 to 2015.

cases, the result of the assay shows flat or slowly rising real-time RT-PCR amplification curves. The delayed $C_T$ values for M gene detection is caused by a mismatch of the probe or primer pair sequences or of both. Thus, there is need for a separate real-time RT-PCR assay (InfA F and InfA R) for Influenza A virus detection. In this study, the delayed $C_T$ values obtained when the MAF and MAR primers and MA probe were used (S2 Table) could be used to distinguish Influenza A pdm H1N1 and H3N2 serotypes. Specifically, Influenza A pdm H1N1 virus could be identified when the $C_T$ values obtained based on the use of the MAF and MAR primers and MA probe were very similar to those obtained when the InfAF and InfAR primers and InfA probe were used (S1 Table). Conversely, Influenza A H3N2 could only be identified when the $C_T$ values obtained based on the use of MAF and MAR primers and the MA probe were much greater than those detected based on the use of InfAF and InfAR primers and InfA probe or were even undetectable (S2 Table). Our combined multiplex real-time RT-PCR offered the possibility to rapidly detect both Influenza A and B viruses and distinguish Influenza A pdm H1N1 and H3N2 subtypes within 2.5 h. The total test duration did not include the time for viral RNA extraction. During the influenza season, human Influenza B virus and Influenza A H3N2 and pdm H1N1 viruses have been identified as pathogens that cause respiratory diseases. Therefore, this multiplex real-time RT-PCR, including three different groups of primers and probes could be used extensively. Overall, a high correlation was observed

between the diagnostic performance of the traditional and multiplex Influenza A virus detection methods, and Influenza A H3N2 (subtype 558) and pdm H1N1 (subtype 401) viruses were successfully distinguished in this study using our multiplex system.

Every influenza virus genome segment evolves differently owing to differences in selection pressures [22, 23]. Comparisons of the rate differences among various segments have indicated that the substitution rates of surface proteins, HA, and NA are higher than those of internal viral proteins [22–24]. However, between 1992 and 2005, it was observed that the M and NS segments of Influenza A H3N2 virus evolved at a rate of $5.2 \times 10^{-3}$ nucleotide substitutions per site per year. This is comparable to the rate of $5.72 \times 10^{-3}$ nucleotide substitutions per site per year for the HA gene [11]. Given the high mutation rates of HA and NA, M segments were used to identify and the subtypes of Influenza A viruses, and constant real-time gene sequencing was required to maintain a high level of sensitivity in the detection assays [8]. Therefore, a rapid, highly sensitive, and specific system must be developed for detecting Influenza A and B viruses [8, 10]. In this study, we established a multiplex platform for the differential detection of the M gene of Influenza A and B viruses based on real-time RT-PCR techniques.

Phylogenetic analysis of the partial HA gene sequence has shown a higher evolution rate for the HA gene from influenza pdmH1N1 and H3N2 viruses in different influenza seasons [20]. Further, it has been observed that nucleotide mutations induce changes in amino acid sequences and help circulating influenza A viruses withstand challenges from the immune system or antiviral drugs [25]. The new clade of influenza A pdm H1N1 and H3N2 viruses was clustered within a period to 6–18 months. Clade 12.3, the new clade of Influenza A pdm H1N1 virus formed in 2012, while clade 11.2 formed between 2013 and 2015 (**Fig 4A**). The new clade TW2014–2015 of influenza H3N2 formed in 2014 and 2015 (**Fig 4B**). Traditional HA typing primers may yield false-negative results because of the higher evolution rates in the HA gene in influenza pdmH1N1 and H3N2 viruses. However, the M protein gene of influenza A viruses has a lower mutation rate and is often used to detect influenza A in many experimental laboratories [10, 25].

In a recent study, [26], a multiplex real-time RT-PCR was developed for the simultaneous detection of the four most common avian respiratory viruses: avian influenza virus, infectious bronchitis virus, Newcastle disease virus, and infectious laryngotracheitis virus. In this previous study various primers and probes for these viruses, which were encompassed into one tube for simultaneous detection, were used. This approach is similar to ours except that our multiplex assay was designed for the simultaneous detection of Influenza B virus or pdm H1N1 and H3N2 viruses, the two most prevalent strains of Influenza A virus. Certain commercial kits, such as FilmArray multiplex PCR [27] apply nucleic acid amplification test-based methods to detect multiple respiratory pathogens in a single test. However, unlike the results of our multiplex assay, reporting $C_T$ values, the results obtained based on these commercial kits show only positive or negative diagnosis.

Coronavirus disease 2019 (COVID-19), caused by severe acute respiratory syndrome coronavirus 2 (SARS-CoV-2), has remained prevalent since 2019. Thus, COVID-19 is an ongoing public health emergency that poses a great challenge to global healthcare systems [28]. Similar to the combined detection of Influenza A and B viruses undertaken in this study, measures to reduce its spread critically depend on the timely and accurate identification of individuals with suspected infection [8]. Further, as real-time RT-PCR is considered the gold standard for detecting SARS-CoV-2, the multiplex PCR method developed in this study could be modified for the simultaneous detection of Influenza A H3N2, pdm H1N1, Influenza B, and SARS-CoV-2 viruses. It could also serve as an important infection control and management tool in the upcoming flu season in this period of the COVID-19 pandemic [8, 28, 29].

## Conclusions

In this study, we developed a multiplex real-time RT-PCR method for the highly efficient and accurate detection and differentiation of Influenza A and B viruses, and subtyping of Influenza A virus within 2.5 h. This multiplex RT-PCR assay offers the possibility to rapidly diagnose patients and can contribute to the improvement of patient outcomes and medical cost reduction.

## Supporting information

**S1 Table. $C_T$ values of Influenza A pdm H1N1 detected from representative clinical samples and viral isolations by the two real-time RT-PCR assays.**
(DOCX)

**S2 Table. The $C_T$ value of Influenza A H3N2 detected from representative clinical samples and viral isolations by the two real-time RT-PCR assays.**
(DOCX)

**S3 Table. Raw data of Table 3.**
(DOCX)

## Acknowledgments

The authors are grateful to Professor Yu Ru Kou for his valuable suggestions in the preparation of this manuscript and Editage for assisting with language editing.

## Author Contributions

**Conceptualization:** Hui-Hua Yang, Li-Kuang Chen.

**Data curation:** Hui-Hua Yang, I-Tsong Huang, Ren-Chieh Wu.

**Formal analysis:** Hui-Hua Yang, I-Tsong Huang.

**Funding acquisition:** Hui-Hua Yang, I-Tsong Huang, Li-Kuang Chen.

**Investigation:** Hui-Hua Yang, I-Tsong Huang, Ren-Chieh Wu, Li-Kuang Chen.

**Methodology:** Hui-Hua Yang, I-Tsong Huang.

**Project administration:** Hui-Hua Yang, Li-Kuang Chen.

**Resources:** Hui-Hua Yang, Ren-Chieh Wu, Li-Kuang Chen.

**Supervision:** Hui-Hua Yang, Li-Kuang Chen.

**Validation:** Hui-Hua Yang.

**Visualization:** I-Tsong Huang.

**Writing – original draft:** Hui-Hua Yang, I-Tsong Huang.

**Writing – review & editing:** Hui-Hua Yang, I-Tsong Huang, Ren-Chieh Wu, Li-Kuang Chen.

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
