## [Decision Letter · Decision Letter 0]

23 Aug 2022

PONE-D-22-10495A highly efficient and accurate method for detecting and subtyping Influenza A pdm H1N1 and H3N2 viruses with newly emerging mutations in the matrix gene in Eastern TaiwanPLOS ONE

Dear Dr. Chen,

Thank you for submitting your manuscript to PLOS ONE. After careful consideration, we feel that it has merit but does not fully meet PLOS ONE’s publication criteria as it currently stands. Therefore, we invite you to submit a revised version of the manuscript that addresses the points raised during the review process. Your manuscript has been reviewed by two peer-reviewers and their reports are appended below.  The reviewers comment that aspects of the study need further explanation, clarification, or review in the terminology used. In addition, the reviewers comment that some of the methodology requires a bit more detail, and that the study does not present the required statistics to assess and compare the sensitivity of the primer sets and probes used in this study.  Can you please address the comments raised by the reviewers?

We look forward to receiving your revised manuscript.

Kind regards,

Maria Elisabeth Johanna Zalm, Ph.D

Editorial Office

PLOS ONE

Journal Requirements:

This work was supported by Grants from the Centers for Disease Control, R.O.C (Taiwan). (HZ099103, HZ100102, HZ101060, CH102055, CW103035).

Reviewers' comments:

Reviewer's Responses to Questions

**Comments to the Author**

1. Is the manuscript technically sound, and do the data support the conclusions?

Reviewer #1: Partly

Reviewer #2: Yes

2. Has the statistical analysis been performed appropriately and rigorously? 

Reviewer #1: No

Reviewer #2: N/A

3. Have the authors made all data underlying the findings in their manuscript fully available?

Reviewer #1: Yes

Reviewer #2: Yes

4. Is the manuscript presented in an intelligible fashion and written in standard English?

Reviewer #1: No

Reviewer #2: Yes

5. Review Comments to the Author

Reviewer #1: The manuscript entitled “ A highly efficient and accurate method for detecting and subtyping Influenza A pdm H1N1 and H3N2 viruses with newly emerging mutations in the matrix gene in Eastern Taiwan" developed an updated and faster method for detection and subtyping of influenza viruses from clinical samples using real-time PCR.

The manuscript highlight the importance of regular checking and update of the sequence of primers and probes used for routine diagnosis of influenza from clinical samples however there are a few aspects that require attention before it is suitable for publication :

1- The manuscript has to be revised for English language , there is several words used which is giving another meaning and impression like :

line 56 : Influenza A virus strains can "Stably" infect humans... : Stable is not the right word

Line 91: Primers and Probes failing to "Adhere" : Bind is much better to use here

Line 319 : Influenza virus genome segment "Develops" , "evolve" would be the correct word.

and so on , the mistakes with the English Language is repeated in several sections of the MS so i would recommend professional English editing before publication.

2-Line :152 , Please give more details regarding your positive and negative controls for both real-time PCR and conventional PCR.

3- You used the Term "RNA sequencing" , this is somehow misleading since you only made sequencing of the viral cDNA and not the RNA.

4- No statistics made , Statistical analysis has to be made to address the significance and compare the sensitivity of the 2 sets of primers and probes for better conclusion.

Reviewer #2: The manuscript entitled” A highly efficient and accurate method for detecting and subtyping Influenza A pdm H1N1 and H3N2 viruses with newly emerging mutations in the matrix gene in Eastern Taiwan” (Manuscript Number: PONE-D-22-10495) was reviewed, the number of samples were studied in this manuscript are very valuable. These results can provide information on clinically correct medication and vaccine design, but some cases not clear such as:

1) As mentioned in this paper the aim of this project was avoid the false negatives caused using the WHO-recommended primers and probes for detecting influenza A and B viruses, and to shorten total time needed to differentiate influenza A and B. Authors must be clarified this time under 2.5 hours, is include the time of viral RNA extraction? If is include, how many samples?

2) Considering that there are currently rapid one-step commercial kits and some papers, for detecting several respiratory viruses at the same time, authors must be comprised their methods with these commercial kits in the discussion section of the article.

such as: A multiplex real-time RT-PCR for simultaneous detection of four most common avian respiratory viruses

6. PLOS authors have the option to publish the peer review history of their article (what does this mean?). If published, this will include your full peer review and any attached files.

Reviewer #1: No

Reviewer #2: No

---

## [Author Response · Author response to Decision Letter 0]

19 Oct 2022

Responses to Editors and Reviewers (PONE-D-22-10495)

We would like to thank the editor and reviewers for their extensive assessment of our manuscript, and for important and helpful comments and suggestions. We have taken all the remarks into account, in a manner that is described in detail below together with our responses to comments. We have responded to all the editor’s and reviewer’s comments in a point-by-point fashion and have revised the manuscript accordingly. The revised portions are indicated by “Track Changes”. An “unmarked” clean version of revised manuscript has also been provided. We think that, following these suggestions, our manuscript has gained in clarity and hope that the changes made will be considered satisfactory.

One additional remark is that please update our Funding Statement in the online submission, which should read as: “This work was supported by Grants from the Centers for Disease Control, R.O.C (Taiwan). (HZ099103, HZ100102, HZ101060, CH102055, CW103035).”

Responses to Editor:

Comment 1: Please include the following items when submitting your revised manuscript.

Response: We have provided a rebuttal letter that responds to the academic editor and reviewers in a point-by-point fashion. We have uploaded a marked-up copy of our revised manuscript that highlights changes made to the original version labeled as 'Revised Manuscript with Track Changes'. We have also uploaded an unmarked version without tracked changes labeled as 'Manuscript'.

Comment 2: Please ensure that your manuscript meets PLOS ONE's style requirements, including those for file naming.

Response: We have ensured that our manuscript meets the style requirements of PLOS ONE.

Comment 3: If the need for consent was waived by the ethics committee, please include this information.

Response: We have provided a statement to declare that informed consent was waived according to the institutional guidelines (line 126, unmarked version).

Comment 4: We note that the grant information you provided in the ‘Funding Information’ and ‘Financial Disclosure’ sections do not match.

Response: We have corrected this mismatching.

Comment 5: Please remove any funding-related text from the manuscript and let us know how you would like to update your Funding Statement. Please include your amended statements within your cover letter; we will change the online submission form on your behalf.

Response: We have removed the funding-related text from the manuscript. Also, please update our Funding Statement in the online submission, which should read as: “This work was supported by Grants from the Centers for Disease Control, R.O.C (Taiwan). (HZ099103, HZ100102, HZ101060, CH102055, CW103035).”

 

Responses to Reviewer 1:

Comment 1: The manuscript has to be revised for English language, there is several words used which is giving another meaning and impression like:

(a) line 56 : Influenza A virus strains can "Stably" infect humans... : Stable is not the right word

Response to (a): We have deleted the word “stably” (line 61, unmarked version).

(b) Line 91: Primers and Probes failing to "Adhere" : Bind is much better to use here

Response to (b): Thank you. We have replaced the word by “bind” (line 92, unmarked version).

(c) Line 319: Influenza virus genome segment "Develops" , "evolve" would be the correct word.

Response to (c): Thank you. We have replaced the word by “evolve” (line 329, unmarked version).

(d) the mistakes with the English Language is repeated in several sections of the MS so i would recommend professional English editing before publication.

Response to (d): In response to the recommendation from the reviewer, we have asked help to improve the language of our revised manuscript by a professional English editor (KGSupport). The certificate of this English editing service is provided at the end of this document. 

Comment 2: Line :152 , Please give more details regarding your positive and negative controls for both real-time PCR and conventional PCR.

Response: We thank the reviewer for reminding us this important issue. In response to this comment, we have now provided details regarding our positive and negative controls for both real-time PCR and conventional PCR (lines 170-173, unmarked version). The statement reads as: “We used RNA extracted from Influenza A/New Caledonia/20/99(H1N1), Influenza A/Wisconsin/67/05(H3N2) and Influenza B Florida/07/04 as positive controls and sterile water as the negative control in each run.”

Comment 3: You used the Term "RNA sequencing" , this is somehow misleading since you only made sequencing of the viral cDNA and not the RNA.

Response: We thank the reviewer for this suggestion. The misleading term has now been replaced by “viral cDNA sequencing” (lines 193 and 194, unmarked version).

Comment 4: No statistics made, Statistical analysis has to be made to address the significance and compare the sensitivity of the 2 sets of primers and probes for better conclusion.

Response: We thank the reviewer for this excellent suggestion. In response to this comment, we have additionally provided data of analytical sensitivity of the conventional singleplex and our multiplex assays analyzed by standard curves of CT values. For comparisons of CT values and Pearson correlation coefficients between the conventional singleplex and our multiplex assays, we have used independent sample t-tests and no significances were detected. Thus, we have made the following revisions:

(1) This additional set of data is presented as Table 5 (page 29, unmarked version). 

(2) The results of these analyses have been reported in the Result section (lines 248-254 , unmarked version). These statements read as:” Table 5 shows the data of the analytical sensitivity of the multiplex assays developed in this study and the traditional singleplex assays. The CT values for each standard concentration obtained from the use of MAF/R, MBF/R, and InfAF/R in the multiplex assays did not significantly differ from those in the singleplex assays. Further analyses of standard curves of CT values revealed that the correlation coefficients, slopes, and amplification efficiencies of these two types of assays had no significant differences.”

(3) The results of these analyses have been reported in the abstract (lines 52-53, unmarked version) and the statement reads as: “The analytical sensitivity of this multiplex RT-PCR assay was as good as that of the conventional singleplex assay.”

(4) A paragraph of statistical analysis has been added to the revised manuscript (lines 224-230, unmarked version).

We sincerely hope that the reviewer could approve our explanations.

 

Responses to Reviewer 2:

Comment 1: As mentioned in this paper the aim of this project was avoid the false negatives caused using the WHO-recommended primers and probes for detecting influenza A and B viruses, and to shorten total time needed to differentiate influenza A and B. Authors must be clarified this time under 2.5 hours, is include the time of viral RNA extraction? If is include, how many samples?

Response: We thank the reviewer for reminding us this important issue. The time of viral RNA extraction was not included in the test duration of 2.5 hours, which was indicated only for the multiplex RT-PCR assay. In response to the reviewer’s comment, we have added a statement in the revised manuscript to clarify this issue (lines 370-371, unmarked version). The statement reads as; “The total test duration did not include the time of viral RNA extraction.”

Comment 2: Considering that there are currently rapid one-step commercial kits and some papers, for detecting several respiratory viruses at the same time, authors must be comprised their methods with these commercial kits in the discussion section of the article. such as: A multiplex real-time RT-PCR for simultaneous detection of four most common avian respiratory viruses

Response: We thank the reviewer for providing the valuable reference, which has been added to our reference list (ref. 25). In response to this comment, we have added a paragraph (lines 378-385, unmarked version) to discuss the methods of that work (ref. 25). The paragraph reads as: “A recent work [25] developed a multiplex real-time RT-PCR for the simultaneous detection of the four most common avian respiratory viruses: avian influenza virus, infectious bronchitis virus, Newcastle disease virus, and infectious laryngotracheitis virus. They used various primers and probes for these viruses, which were encompassed into one tube for simultaneous detections. Their approach was similar to ours except that our multiplex assay was designed for the simultaneous detection of Influenza B virus or pdm H1N1 and H3N2, the two most prevalent strains of Influenza A.”

We have also added statements (lines 385-388, unmarked version) to discuss the methods of certain commercial kits and an additional reference (ref. 26) was added to the list. These statements read as: “Certain commercial kits such as FilmArray multiplex PCR [26] apply nucleic acid amplification test-based methods to detect multiple respiratory pathogens in a single test. However, unlike the results of our multiplex assay reporting CT values, the results of these commercial kits show only positive or negative diagnosis.”

We sincerely hope that the reviewer could approve our explanations.

---

## [Decision Letter · Decision Letter 1]

13 Dec 2022

PONE-D-22-10495R1A highly efficient and accurate method of detecting and subtyping Influenza A pdm H1N1 and H3N2 viruses with newly emerging mutations in the matrix gene in Eastern TaiwanPLOS ONE

Dear Dr. Chen,

Thank you for submitting your manuscript to PLOS ONE. After careful consideration, we feel that it has merit but does not fully meet PLOS ONE’s publication criteria as it currently stands. Therefore, we invite you to submit a revised version of the manuscript that addresses the points raised during the review process.

While the reviewer agreed that the revised version showed improvement, there are still several issues that needs to be addressed.  The authors established a multiplex RT-PCR assay for rapid detection and distinguishing pdm H1N1 and H3N2 in eastern Taiwan. The authors combined WHO and Taiwan CDC suggested primers/probes in the multiplex PCR which mitigated the low detection efficiency of M-mutant H3N2 viruses (evolved and appeared after 2011) using WHO suggested primers/probes set.  The assay was verified using 5709 clinical specimens. While not mentioned in the abstract and introduction, the authors did phylogenetic analysis of 94 samples (44 pdm H1N1 and 50 H3N2). I am at a lost on the intention of the phylogenetic analysis, which needs to be mentioned in abstracts and introduction to strengthen the rationale of the study.

In addition, the quality of the language still needs to be improved. We suggest you thoroughly copyedit your manuscript for language usage, spelling, and grammar. If you do not know anyone who can help you do this, you may wish to consider employing a professional scientific editing service.

Whilst you may use any professional scientific editing service of your choice, PLOS has partnered with both American Journal Experts (AJE) and Editage to provide discounted services to PLOS authors. Both organizations have experience helping authors meet PLOS guidelines and can provide language editing, translation, manuscript formatting, and figure formatting to ensure your manuscript meets our submission guidelines. To take advantage of our partnership with AJE, visit the AJE website (http://learn.aje.com/plos/) for a 15% discount off AJE services. To take advantage of our partnership with Editage, visit the Editage website (www.editage.com) and enter referral code PLOSEDIT for a 15% discount off Editage services. If the PLOS editorial team finds any language issues in text that either AJE or Editage has edited, the service provider will re-edit the text for free."

Specific comments:

Abstract, line 45 – 46:  This sentence is not relevant. Suggest deletion.Materials and method, line 209:  BLAST please reference “J Mol Biol. 1990 Oct 5;215(3):403-10. doi: 10.1016/S0022-2836(05)80360-2.Basic local alignment search tool S F Altschul 1, W Gish, W Miller, E W Myers, D J Lipman”Results, line 247 – 248:  Suggest moving tables 3 & 4 to supplemental materials.The purpose of the 44 pdm H1N1 and 50 H3N2 samples for phylogenetic analysis.Discussion, line 341 – 344: This sentence is not relevant.  Suggest deletion.Discussion, line 357 – 377:  Suggest moving this paragraph to the beginning of the discussion.Discussion, line 366 – 368:  Please explain how the delayed C_T_ values help distinguishing pdm H1N1 and H3N2. Where is Table 1A?Please submit your revised manuscript by Jan 27 2023 11:59PM. If you will need more time than this to complete your revisions, please reply to this message or contact the journal office at plosone@plos.org. Please include the following items when submitting your revised manuscript:A rebuttal letter that responds to each point raised by the academic editor and reviewer(s). You should upload this letter as a separate file labeled 'Response to Reviewers'.A marked-up copy of your manuscript that highlights changes made to the original version. You should upload this as a separate file labeled 'Revised Manuscript with Track Changes'.An unmarked version of your revised paper without tracked changes. You should upload this as a separate file labeled 'Manuscript'.

We look forward to receiving your revised manuscript.

Kind regards,

Baochuan Lin, Ph.D.

Academic Editor

PLOS ONE

Reviewers' comments:

Reviewer's Responses to Questions

**Comments to the Author**

1. If the authors have adequately addressed your comments raised in a previous round of review and you feel that this manuscript is now acceptable for publication, you may indicate that here to bypass the “Comments to the Author” section, enter your conflict of interest statement in the “Confidential to Editor” section, and submit your "Accept" recommendation.

Reviewer #1: All comments have been addressed

2. Is the manuscript technically sound, and do the data support the conclusions?

Reviewer #1: Yes

3. Has the statistical analysis been performed appropriately and rigorously? 

Reviewer #1: Yes

4. Have the authors made all data underlying the findings in their manuscript fully available?

Reviewer #1: Yes

5. Is the manuscript presented in an intelligible fashion and written in standard English?

Reviewer #1: Yes

6. Review Comments to the Author

Reviewer #1: (No Response)

7. PLOS authors have the option to publish the peer review history of their article (what does this mean?). If published, this will include your full peer review and any attached files.

Reviewer #1: No

---

## [Author Response · Author response to Decision Letter 1]

26 Dec 2022

Responses to Editor (PONE-D-22-10495_R2)

We would like to thank the editor for their extensive assessment of our manuscript, and for important and helpful comments and suggestions. We have taken all the remarks into account, in a manner that is described in detail below together with our responses to comments. We have responded to all the editor’s comments in a point-by-point fashion and have revised the manuscript accordingly. The revised portions are indicated by “Track Changes”. An “unmarked” clean version of revised manuscript has also been provided. We think that, following these suggestions, our manuscript has gained in clarity and hope that the changes made will be considered satisfactory.

Responses to Editor’s Comments:

Major Comments:

Comment 1: While not mentioned in the abstract and introduction, the authors did phylogenetic analysis of 94 samples (44 pdm H1N1 and 50 H3N2). I am at a lost on the intention of the phylogenetic analysis, which needs to be mentioned in abstracts and introduction to strengthen the rationale of the study.

Response: We thank the editor for this excellent suggestion. In response to the suggestion, we have added statements in the abstract and introduction to address this issue. The statements in the abstract read as:” Further, the phylogenetic analyses of our samples revealed that the characteristics of these viruses were different from those reported previously using samples collected during 2012–2013” (lines 54-56). The statements in the introduction section read as:” In this study, we also report the results of the phylogenetic analyses of Influenza A pdm H1N1 and H3N2 viruses associated with the pandemic that occurred during 2012–2013 in Taiwan [14]. Since then, the changes in phylogenetic characteristics of these viruses have not yet been investigated. Therefore, in this study, we further performed phylogenetic analyses for the Influenza A pdm H1N1 and H3N2 viruses that caused the 2011–2015 pandemic” (lines 117-122).

Comment 2: In addition, the quality of the language still needs to be improved. We suggest you thoroughly copyedit your manuscript for language usage, spelling, and grammar. If you do not know anyone who can help you do this, you may wish to consider employing a professional scientific editing service.

Response: In response to the suggestion, our revised manuscript (R2 version) has been professionally edited by Editage for improvements of language; this company is recommended by the journal. The certificate of this English editing service has been uploaded alongside the revised manuscript.

Specific comments:

Comment 1: Abstract, line 45 – 46: This sentence is not relevant. Suggest deletion.

Response: In response to the suggestion, the sentence has been deleted (at the position of line 45).

Comment 2: Materials and method, line 209: BLAST please reference “J Mol Biol. 1990 Oct 5;215(3):403-10. doi: 10.1016/S0022-2836(05)80360-2. Basic local alignment search tool S F Altschul 1, W Gish, W Miller, E W Myers, D J Lipman”

Response: We thank the editor for providing this important reference, which has been added to the reference list and cited (Materials and method, line 233). The numbers of other references in the text have been revised accordingly.

Comment 3: Results, line 247 – 248: Suggest moving tables 3 & 4 to supplemental materials.

Response: In response to the suggestion, these two tables have been moved to supplemental materials (S1 and S2 Tables) (line 273). The number of Table 5 in the text have been changed to Table 3 accordingly (line 274 and line 290).

Comment 4: The purpose of the 44 pdm H1N1 and 50 H3N2 samples for phylogenetic analysis.

Response: In response to this suggestion, we have added statements in the abstract section to address this issue.” Further, the phylogenetic analyses of our samples revealed that the characteristics of these viruses were different from those reported previously using samples collected during 2012–2013” (lines 54-56). The statements in the introduction section read as:” In this study, we also report the results of the phylogenetic analyses of Influenza A pdm H1N1 and H3N2 viruses associated with the pandemic that occurred during 2012–2013 in Taiwan [14]. Since then, the changes in phylogenetic characteristics of these viruses have not yet been investigated. Therefore, in this study, we further performed phylogenetic analyses for the Influenza A pdm H1N1 and H3N2 viruses that caused the 2011–2015 pandemic” (lines 117-122).

Comment 5: Discussion, line 341 – 344: This sentence is not relevant. Suggest deletion.

Response: In response to the suggestion, the sentence has been deleted (line 408).

Comment 6: Discussion, line 357 – 377: Suggest moving this paragraph to the beginning of the discussion.

Response: We thank the editor for the suggestion. This paragraph has been moved to the first paragraph of the discussion (lines 365-394).

Comment 7: Discussion, line 366 – 368: Please explain how the delayed CT values help distinguishing pdm H1N1 and H3N2. Where is Table 1A?

Response: Table 1A was a typo and should be Table 4 in our last R1 version. The Table 1A has been corrected to S2 Table (supplementary materials) in this R2 version. We apologize for the error we made. We have also added statements to explain how the delayed CT values help distinguishing pdm H1N1 and H3N2. These statements read as:” In this study, the delayed CT values obtained when the MAF and MAR primers and MA probe were used (S2 Table) could be used to distinguish Influenza A pdm H1N1 and H3N2 serotypes. Specifically, Influenza A pdm H1N1 virus could be identified when the CT values obtained based on the use of the MAF and MAR primers and MA probe were very similar to those obtained when the InfAF and InfAR primers and InfA probe were used (S1 Table). Conversely, Influenza A H3N2 could only be identified when the CT values obtained based on the use of MAF and MAR primers and the MA probe were much greater than those detected based on the use of InfAF and InfAR primers and InfA probe or were even undetectable (S2 Table)” (lines 375-383).

---

## [Editor Report · Decision Letter 2]

6 Jan 2023

PONE-D-22-10495R2

A highly efficient and accurate method of detecting and subtyping Influenza A pdm H1N1 and H3N2 viruses with newly emerging mutations in the matrix gene in Eastern Taiwan

PLOS ONE

Dear Dr. Chen,

Thank you for submitting your manuscript to PLOS ONE. After careful consideration, we feel that it has merit but does not fully meet PLOS ONE’s publication criteria as it currently stands. Therefore, we invite you to submit a revised version of the manuscript that addresses the points raised during the review process.

The revised manuscript showed significant improvement over previous version. However, there are a few issues that still need to be addressed.

1. Line 83, suggest changing "Therefore, presently,..." to "Presently..."

2. Line 171, delete "of"

3. Line 268, suggest changing "Further, the..." to "The..."

4. Line 268 - 269, It seems that table S1 and S2 only listed representative samples of pdmH1N1, and H3N2, not all the test samples. Please clarify.

5. Line 271, suggest deleting "data"

6. Line 309 - 310, need to rephrase this sentence for clarity. Does the second set of primers and probe remedy the situation mentioned in the previous sentences?

7. Line 369, suggest changing "...sometimes causes..." to "...caused by..."

8. Line 370, suggest deleting "simultaneously applied"

9. Line 371-372, suggest deleting "...with early CT values..."

10. Line 419, suggest deleting "...previous..."

We look forward to receiving your revised manuscript.

Kind regards,

Baochuan Lin, Ph.D.

Academic Editor

PLOS ONE
---

## [Author Response · Author response to Decision Letter 2]

16 Jan 2023

Responses to Editor (PONE-D-22-10495_R3)

We would like to thank the editor for their extensive assessment of our manuscript, and for important and helpful comments and suggestions. We have taken all the remarks into account, in a manner that is described in detail below together with our responses to comments. We have responded to all the editor’s comments in a point-by-point fashion and have revised the manuscript accordingly. The revised portions are indicated by “Track Changes”. An “unmarked” clean version of revised manuscript has also been provided. We think that, following these suggestions, our manuscript has gained in clarity and hope that the changes made will be considered satisfactory.

Responses to Editor’s Comments:

General comment: The revised manuscript showed significant improvement over previous version. However, there are a few issues that still need to be addressed.

Response: We thank the editor for his/her positive feedback and for valuable suggestions to enhance the quality of our work. We also thank the editor for his/her careful reading of our manuscript.

Comment 1: Line 83, suggest changing "Therefore, presently,..." to "Presently..."

Response: This has been changed (Line 83).

Comment 2: Line 171, delete "of"

Response: This has been deleted (Line 171).

Comment 3: Line 268, suggest changing "Further, the..." to "The..."

Response: This has been changed (Line 268).

Comment 4: Line 268 - 269, It seems that table S1 and S2 only listed representative samples of pdmH1N1, and H3N2, not all the test samples. Please clarify.

Response: The editor is correct regarding this issue. Table S1 and S2 only listed representative samples of pdmH1N1, and H3N2, not all the test samples. In response to this suggestion, we have rephrased the statement as:” The threshold cycle (CT) values of some representative tested samples diagnosed …” (Line 268-269). Additionally, the titles of tables S1 and S2 have been changed to “… detected from representative clinical samples…”(Supplemental Materials).

Comment 5: Line 271, suggest deleting "data"

Response: This has been deleted (Line 271).

Comment 6: Line 309 - 310, need to rephrase this sentence for clarity. Does the second set of primers and probe remedy the situation mentioned in the previous sentences?

Response: We thank the reviewer for reminding us this important issue. In response to the suggestion, the statement has been rephrased and reads as:” This situation can be remedied by our second set of primers and probe (InfAF and InfAR primers, and InfA probe), which targeted a more conserved region of the M gene sequence of Influenza A H3N2 virus” (Line 309-311).

Comment 7: Line 369, suggest changing "...sometimes causes..." to "...caused by..."

Response: This has been changed (Line 369).

Comment 8: Line 370, suggest deleting "simultaneously applied"

Response: This has been deleted (Line 370).

Comment 9: Line 371-372, suggest deleting "...with early CT values..."

Response: This has been deleted (Line 372).

Comment 10: Line 419, suggest deleting "...previous..."

Response: This has been deleted (Line 419).

---

## [Editor Report · Decision Letter 3]

2 Mar 2023

A highly efficient and accurate method of detecting and subtyping Influenza A pdm H1N1 and H3N2 viruses with newly emerging mutations in the matrix gene in Eastern Taiwan

PONE-D-22-10495R3

Dear Dr. Chen,

We’re pleased to inform you that your manuscript has been judged scientifically suitable for publication and will be formally accepted for publication once it meets all outstanding technical requirements.

Kind regards,

Baochuan Lin, Ph.D.

Academic Editor

PLOS ONE
---

## [Editor Report · Acceptance letter]

13 Mar 2023

PONE-D-22-10495R3 

A highly efficient and accurate method of detecting and subtyping Influenza A pdm H1N1 and H3N2 viruses with newly emerging mutations in the matrix gene in Eastern Taiwan 

Dear Dr. Chen:

I'm pleased to inform you that your manuscript has been deemed suitable for publication in PLOS ONE. Congratulations! Your manuscript is now with our production department. 

Kind regards, 

on behalf of

Dr. Baochuan Lin 

Academic Editor

PLOS ONE